# Power and sample size for reversible linear mixed models with clustering and longitudinality: GLIMMPSE Version 3

Deborah H. Glueck[1]*, Qian Li[2], Alasdair J. Macleod[3], Elizabeth M. Litkowski[4], Xi Yang[2], Jiang Bian[5], Albert D. Ritzhaupt[6], Max Sommer[6], Natercia Valle[7], Jessica R. Shaw[8], Keith E. Muller[2]

1 Department of Pediatrics, University of Colorado Denver, Aurora, Colorado, United States of America, 2 Department of Health Outcomes and Biomedical Informatics, University of Florida, Gainesville, Florida, United States of America, 3 Applications Directorate, University of Edinburgh, Edinburgh, Scotland, United Kingdom, 4 Michigan Neurosciences Institute, University of Michigan, Ann Arbor, Michigan, United States of America, 5 Regenstrief Institute, Indiana University, Indianapolis, Indiana, United States of America, 6 School of Teaching and Learning, College of Education, University of Florida, Gainesville, Florida, United States of America, 7 Marston Science Library, University of Florida, Gainesville, Florida, United States of America, 8 Department of Biostatistics, Colorado School of Public Health, Anschutz Medical Campus, Aurora, Colorado, United States of America

* Deborah.Glueck@cuanschutz.edu

**Data availability statement:** The software is open source, and copyleft under the GNU public license. The software is freely available at the

## Abstract

GLIMMPSE Version 3 is a free, web-based, open-source software tool, which calculates power and sample size for general linear mixed models with Gaussian errors. The software permits power calculations for clinical trials, randomized experiments, and observational studies with clustering, repeated measures, and both, and almost any testable hypothesis. The software has been supported by five United States National Institutes of Health (NIH) grants, is used for more than 14,000 power or sample size calculations per year, has been cited in almost 500 peer-reviewed manuscripts, and used to design more than 200 million dollars in NIH-funded studies. This release provides several new features. The back end has been refactored in Python. The interface has been simplified, requiring user decisions about only one topic per screen. A new menu improves specification of both between-participant and within-participant hypotheses. A recursive algorithm permits computing covariances for up to ten levels of clustering. An updated Monte Carlo simulation using five new examples with clustering, longitudinality, or both, shows accuracy of the power approximations to within 0.01. Five new examples demonstrate power or sample size calculations for 1) a cluster-randomized trial, 2) a longitudinal study with repeated measures, 3) a multilevel study with a multivariate outcome, 4) a multilevel and longitudinal study, and 5) a complex study with a subgroup factor, repeated measures, and intervention-by-location interaction.

website https://glimmpse.samplesizeshop.org/. We did not upload the code base, as the size of the code precluded upload at the submission site. Code for the front end, back end, and Python libraries are at, respectively, https://github.com/SampleSizeShop/glimmpseV3Front, https://github.com/SampleSizeShop/glimmpseV3Back, and https://github.com/SampleSizeShop/pyGlimmpse. Code for the validation experiment (backend only) is at https://github.com/SampleSizeShop/glimmpseV3back.

**Funding:** Major funding was provided by NIH/NIGMS 9R01GM121081 to the University of Colorado Denver (Dana Dabelea, Deborah H. Glueck, Keith E. Muller MPI), and by NIH/NIGMS 5R25GM111901 to the University of Florida (Deborah H. Glueck, Keith E. Muller MPI). Explanatory material mirrors content developed for NIH/NLM 5G13LM011879, awarded to the University of Florida (Deborah H. Glueck, Keith E. Muller, MPI). Initial funding for a previous version of GLIMMPSE was provided by NIH/NIDCR 1 R01 DE020832-01A1 to the University of Florida (Keith E. Muller, PI) and by an American Recovery and Re-investment Act supplement (3K07CA088811-06S) for NIH/NCI grant K07CA088811, awarded to the University of Colorado Denver (Deborah H. Glueck, PI). The funders had no role in study design, data collection and analysis, decision to publish, or preparation of the manuscript.

**Competing interests:** The authors have declared that no competing interests exist.

# 1 Introduction

## 1.1 Non-technical overview

Power and sample size analyses for simple designs and hypotheses are well known. For example, scientists can easily compute power for a *t*-test, when studying differences between the means of normally distributed outcomes drawn two populations with similar variances. Often, however, both observational studies and randomized experiments, like clinical trials, use more complex designs. Scientists often study repeated measures of outcomes across time. Sometimes studies involve clusters. *Clusters* are a group, with membership determined by common exposures, experiences, backgrounds, or interactions between group members [1]. Societal hierarchies, such as schools, can create multiple levels of clustering, with student nested within classroom, and classroom nested within school. Designs with multiple levels of clustering are called *multilevel* studies. Sophisticated power and sample size programs are required to compute power and sample size for studies with repeated measures, clustering, or both repeated measures and clustering.

GLIMMPSE Version 3 is an open source, point-and-click, power and sample size software program for studies with repeated measures, clustering, or both repeated measures and clustering. GLIMMPSE Version 3 provides power and sample size calculations for designs with continuous outcomes and normally distributed errors. Many commonly analyzed outcomes are continuous and have normally distributed errors. Examples include outcomes such as cholesterol, blood pressure, or weight. Results from GLIMMPSE Version 3 will be useful to scientists conducting observational studies, as well as randomized clinical trials and lab experiments.

In the manuscript, we follow a commonly used convention for assessing the number of levels in the design [1–3]. An *independent sampling unit* is an element in a study design which is statistically independent from all other units of the same sort [4]. An independent sampling unit is considered a level in the discussion of multilevel designs. In turn, studies with one level of clustering, such as group- or cluster-randomized trials, are called two-level designs. Three-level designs are studies with, for example, patients nested within practices, and practices nested within health systems (the independent sampling unit). In a four-level design, scientists might study rats nested within a cage, cages nested within shelves, and shelves nested within rodent colonies (the independent sampling unit).

## 1.2 GLIMMPSE

GLIMMPSE provides power and sample size calculations for many study designs which can be analyzed with the general linear multivariate model or the general linear mixed model. Examples include analysis of variance (ANOVA), analysis of covariance (ANCOVA), multivariate analysis of variance (MANOVA), and multivariate analysis of covariance (MANCOVA). The package also can compute power and sample size for studies with clusters or groups, such as group-randomized trials, or studies with multiple levels of nested clusters.

GLIMMPSE is free and open source, and copyleft under the GNU public license. A point-and-click interface guides users. No programming expertise is assumed.

The power computations use exact and approximate results described in Muller and Peterson [5], Muller and Barton [6], Muller et al. [7], Glueck and Muller [8], Muller and Stewart [4], Muller et al. [9], and Chi et al. [10]. Details appear in cited papers, and in the appendix of Kreidler et al. [11].

## 1.3 Novel features in GLIMMPSE Version 3

GLIMMPSE Version 3 adds three new features. First, the software can compute power and sample size for studies with clusters of up to 10,000 elements. Second, the interface to enter within independent sampling unit hypotheses, and between independent sampling unit hypotheses has been updated and clarified. Third, the software now uses a cluster covariance matrix described by Longford [12], in contrast to the Kronecker product cluster covariance matrix used in the last GLIMMPSE release [11]. In tandem with the new features, the low-level libraries have been refactored in Python, and the interface updated.

## 1.4 Literature review

Table 1 includes packages that provide power and sample size for designs with clustering, repeated measures, or both clustering and repeated measures. We excluded packages from the table which provide power for single designs and single hypotheses, and packages which are no longer maintained.

Based on these inclusion and exclusion criteria, several packages were excluded from Table 1. We briefly describe each excluded package. Basagaña and Spiegelman [13,14] describe free software for a range of epidemiological studies. Zhang and Zhang [15] list several R power and sample size modules for designing clinical and cluster randomized trials. Web-based power tools developed by Lenth [16] and Schoenfeld [17] provide power calculations for specific linear models such as t tests, ANOVA, and cross-over studies. Specific examples of packages limited to two or fewer levels of nesting included the Research Methods Resources of the National Institutes of Health (accessed March 19, 2025). The Mplus package [18] allows specifying a data analysis and conducting a simulation for a specific design in order to assess power. Finch and Bolin [19] described how to allow up to three more levels of nesting and up to two levels of clustering in Mplus. Iddi and Donohue [20] reviewed power and sample size modules for longitudinal models in R, and described the features of their module LongPower.

**Table 1. Review of software for fairly general designs with clustering, repeated measures, or both. A checkmark indicates that a feature is available. A blank space indicates that the feature is not available.**

| Features | CRT-Power [25] | SPSS [26] | NQuery [27] | PASS [28] | SAS [29] | Stata [30] | Power & Precision [31] | Powerlib [32] | Powerup [33] | SPA-ML [3] | OD [34] | GLIMMPSE [11] |
|---|---|---|---|---|---|---|---|---|---|---|---|---|
| Free | | | | | | | | ✓ | ✓ | ✓ | ✓ | ✓ |
| Open Source | | | | | | | | ✓ | ✓ | | | ✓ |
| Small $N$ Accuracy | | ✓ | | ✓ | ✓ | | | ✓ | | | | ✓ |
| Graphical Interface | ✓ | | ✓ | ✓ | | | ✓ | | ✓ | ✓ | ✓ | ✓ |
| Multiple Outcomes | | ✓ | | ✓ | ✓ | | | ✓ | | | | ✓ |
| Repeated Measures | | ✓ | ✓ | ✓ | ✓ | | | ✓ | | | ✓ | ✓ |
| Multilevel & Repeated | | | | | | | | ✓ | | | | ✓ |
| Covariates | ✓ | | | | | | | | ✓ | | | ✓ |
| Unequal Clusters | ✓ | | | ✓ | | | ✓ | | ✓ | ✓ | | |
| Subgroup Factor | | | | | | | | ✓ | | | | ✓ |
| Composite Outcome | | | | | | | | ✓ | | | | ✓ |
| Clustering Levels | 3 | 1 | 2 | 3 | 1 | 1 | 1 | >3 | 3 | 3 | 3 | 10 |

LongPower allows various complications that can occur in clinical trials. However, the range of designs is limited, and the statistical methods are all large sample approximations, which are of less accuracy with designs with few clusters, which are very common in many fields.

The package clusterPower [21,22] was removed from the CRAN repository, and archived 2023-10-10. The rationale for the removal was that check problems were not corrected on time. This removal is documented at https://cran.r-project.org/web/packages/clusterPower/index.html, accessed 05/28/2025. A GitRepo still exists at https://github.com/Kenkleinman/clusterPower, but shows a last update at 01/28/2021.

For calculations in G*Power [23,24], a user must specify four of five inputs (the significance level, the "effect size," the number of groups, power, and the number of observations) in order to compute the fifth. The software allows a range of $T$, $F$, and $\chi^2$ tests. "Repeated measures" are treated as a 1 or 2 way design, assuming compound symmetry, and using the uncorrected univariate approach to repeated measures. Compound symmetry is a covariance structure with equal correlation and equal variances, based on the assumption of exchangeability within a cluster. Multivariate settings are not allowed.

None of the packages listed in Table 1 have all the functionality of GLIMMPSE Version 3. CRT Power has the greatest flexibility, but cannot handle repeated measures of the outcome variables, and uses approximate power calculations [25]. PowerUp only allows two treatments [33]. Optimal Design assumes compound symmetric or independent covariance models, even for longitudinal repeated measures [34]. Several packages only allow three levels of nesting. Examples include the National Institutes of Health Research Resource Methods [35] sample size calculators for parallel group-randomized trials, individually randomized trials, and stepped-wedge group-randomized trials. SPA-ML has a limited number of hypotheses, repeated measures, cluster sizes, and number and form of predictors, and uses $z$-approximations for power [3].

GLIMMPSE Version 3 has unique capabilities which no other software package has, including the following:

- GLIMMPSE allows for complex covariance matrices, including unstructured covariances for longitudinal repeated measures;
- GLIMMPSE permits calculations for up to 10 levels of clustering at each level;
- GLIMMPSE is extremely accurate even in small samples;
- GLIMMPSE allows computing power and sample size for almost any hypothesis in many mixed models. This capability permits scientists to compute power and sample size for designs with multiple levels of clustering, repeated measures, or both;
- GLIMMPSE allows saving and re-uploading designs. The resulting files can be published for reproducibility; and
- GLIMMPSE is fast, taking seconds for each calculation.

## 1.5 Organization of the manuscript

The remainder of the manuscript is organized as follows. Sect 2 gives mathematical detail useful for statisticians. Sect 3 describes the updates in the new version, including mathematical details about covariance structures, and algorithmic approaches. Both Sects 2 and 3 may be skipped by readers not interested in technical or mathematical details, without interrupting the narrative flow. Sect 4 has five example power or sample size analyses, of increasing complexity. Sect 5 compares the accuracy of the results to a Monte Carlo simulation. Sect 6 provides details on the number of citers of the software, overall, and per year, and the total

funding of United States National Institutes of Health grants which have used the software. Finally, Sect 7 summarizes the findings.

## 2 The reversible general linear mixed model

### 2.1 Model

*Overview.* A previous publication by Kreidler et al. [11] showed how GLIMMPSE Version 2 provided power calculations for the general linear multivariate model. GLIMMPSE Version 3 retains the functionality, providing power and sample size calculations for the general linear multivariate model. In addition, GLIMMPSE Version 3 provides power and sample size calculations for the reversible general linear mixed model described by Chi et al. [10], a generalization of the balanced general linear mixed model [9,36,37]. Notation and definitions follow those of Sect 6.7, in the book by Muller and Stewart [4].

*Power equivalence.* Power for a test of fixed effects in a reversible general linear mixed model can be computed by transforming the reversible model into a multivariate model [10], as shown below, and using multivariate power results [4,36]. The mixed model Wald test can be transformed into a Hotelling-Lawley trace statistic for the multivariate model [38], as shown below. Using the Kenward and Roger [39] null case approximation for the mixed model Wald statistic provides similar distributional results for the Hotelling-Lawley trace statistic for the multivariate model.

*Sufficient conditions for reversibility.* Not all general linear mixed models are reversible. A general linear mixed model is reversible, only if it satisfies the following three conditions described by Chi *et al.* [10]. First, there must be complete data. This means that there are no missing predictors or outcome variables. Second, covariates have to have the same value for all measures on an independent sampling unit. Finally, the error covariance is assumed to be the same for each independent sampling unit. For a reversible model, all tests of within independent sampling unit hypotheses, between independent sampling unit hypotheses, or between-by-within independent sampling unit hypotheses are testable.

In plain language, this means that GLIMMPSE Version 3 will compute power and sample size for studies with clustering, if every cluster has the same size. The assumptions mean that GLIMMPSE Version 3 will compute power and sample size for longitudinal, spatial or multivariate studies, if there is the same timing, spacing, or variables for all independent sampling units, and no missing data.

Ringham et al. [36] gave examples of reversible linear mixed models. They described a longitudinal study, a group-randomized trial, a three-level group-randomized trial, and a study with multivariate outcomes. In the longitudinal study, investigators plan to study the shape of the weight gain curve of women during pregnancy. Each woman will be measured nine times on the first day of each month of gestation. If an intercept is used as the predictor, the coefficients of the model represent the average weight at each time point measured. In the group-randomized trial, investigators plan to randomize entire workplaces to one of two alcohol reduction programs. Each workplace has the same number of workers. The outcome is the self-reported amount of alcohol ingested by each worker in each workplace in the week following the intervention. When indicator variables for the intervention group are used as predictors, the coefficients for the model represent the average intake across workplaces and workers for each intervention. In the study with multiple outcomes, three oral cancer biomarkers will be measured for each participant. Some of the participants have diagnosed cancer of the oral cavity or pharynx (cases), and some have no diagnoses of oral cancer (controls). Using indicator variables for cases or controls yields model coefficients which are the average values of each biomarker for the cases and the controls.

In practice, in each of the designs described, missing data could occur. In the longitudinal study, women might not come in each month for measures, or might not receive one of the nine measurements on the day that they come in. In the group randomized trial, the clusters might be of different sizes. In the oral cancer study, some participants might have only one or two biomarkers measured, instead of all three. With missing data, or unequal cluster sizes, the models are no longer reversible. In each of these cases, power computed for a reversible model assuming no missing data will be higher than the true power.

GLIMMPSE provides power calculations for reversible mixed models with no missing data. Some rule of thumb approximations for missing data appear in the Discussion. For users who wish to use an analytic approach to account for missing data, Kreidler et al. [40] reviews the literature for power and sample size calculations for designs with missing data, different cluster sizes, or both. In addition, Kreidler et al. [40] derives a non-central $F$ approximation for the Wald test with Kenward and Roger [39] degrees of freedom, and provides an open-source package which allows power calculations.

*The general linear multivariate model and hypothesis.*
The multivariate general linear model is given by

$$Y = X_M B_M + E \, , \tag{1}$$

with $N$ rows corresponding to independent sampling units, $p$ columns in $Y$, $B_M$ and $E$ and $q$ columns in $X_M$. The distributional assumption is that $\text{row}_i \left( E \right)' \sim \mathcal{N}_p \left( 0, \Sigma_M \right)$. The secondary parameter matrix $\Theta_M = C_M B_M U$ is defined in terms of between independent sampling unit contrast matrix $C_M$ and within independent sampling unit contrast matrix $U$. The multivariate general linear hypothesis under the null is

$$H_{0M} : \ C_M B_M U = \Theta_{0M} \, , \tag{2}$$

with $\Theta_{0M}$ a fixed, known and conforming matrix of constants.

*Inputs for power analysis.* Under the assumption that $X_M$ is fixed and of rank $q$ and known without appreciable error, the power analysis can be specified using six matrices: $X_M$, $\Sigma_M$, $C_M$, $B_M$, $U$ and $\Theta_{0M}$. One must also fix the type I error rate, $\alpha$. To simplify calculations, all $X_M$ are defined to have full rank of $q$.

*The general linear mixed model for a single independent sampling unit.* For independent sampling unit $i \in \{1, \dots, N\}$ [4, Sect 6.7], the mixed model includes the $(p_i \times 1)$ matrix of outcomes, $y_i$; the known $(p_i \times q_m)$ matrix of fixed-effect predictors, $X_{mi}$; the $(q_m \times 1)$ matrix of known fixed-effect parameters, $\beta_m$; the $(p_i \times r)$ matrix of fixed and known random-effect predictors, $Z_i$; the $(r \times 1)$ matrix of unknown random-effect values, $d_i$; and the $(p_i \times 1)$ matrix of unknown errors, $e_{mi}$. The independent sampling unit specific mixed model is given by

$$y_i = X_{mi} \beta_m + Z_i d_i + e_{mi} \, , \tag{3}$$

with the distributional assumption of

$$\begin{bmatrix} d_i \\ e_{mi} \end{bmatrix} \sim \mathcal{N}_{r+p_i} \left( \begin{bmatrix} 0 \\ 0 \end{bmatrix}, \begin{bmatrix} \Sigma_{d_i} & 0 \\ 0 & \Sigma_{e_i} \end{bmatrix} \right) \, . \tag{4}$$

The distributional assumptions imply that $\mathbb{E} \left( y_i \right) = X_{mi} \beta_m$ and $\mathcal{V} \left( y_i \right) = Z_i \Sigma_d Z_i' + \Sigma_{e_{mi}}$.

*The model for all independent sampling units.* Equations for the model for all independent sampling units follow. For the entire study sample, one can construct stacked matrices, so that

$$\boldsymbol{y}_m = \begin{bmatrix} \boldsymbol{y}_1 \\ \boldsymbol{y}_2 \\ \vdots \\ \boldsymbol{y}_N \end{bmatrix} , \tag{5}$$

$$\boldsymbol{X}_m = \begin{bmatrix} \boldsymbol{X}_1 \\ \boldsymbol{X}_2 \\ \vdots \\ \boldsymbol{X}_N \end{bmatrix} , \tag{6}$$

$$\boldsymbol{Z} = \begin{bmatrix} \boldsymbol{Z}_1 \\ \boldsymbol{Z}_2 \\ \vdots \\ \boldsymbol{Z}_N \end{bmatrix} , \tag{7}$$

$$\boldsymbol{d} = \begin{bmatrix} \boldsymbol{d}_1 \\ \boldsymbol{d}_2 \\ \vdots \\ \boldsymbol{d}_N \end{bmatrix} , \tag{8}$$

and

$$\boldsymbol{e}_m = \begin{bmatrix} \boldsymbol{e}_1 \\ \boldsymbol{e}_2 \\ \vdots \\ \boldsymbol{e}_N \end{bmatrix} . \tag{9}$$

The general linear mixed model is given by

$$\boldsymbol{y}_m = \boldsymbol{X}_m \beta_m + \boldsymbol{Z}\boldsymbol{d} + \boldsymbol{e}_m . \tag{10}$$

*The population-average mixed model.* For independent sampling unit $i$, the "population-average" form of a mixed model is

$$\boldsymbol{y}_i = \boldsymbol{X}_{mi} \beta_m + \boldsymbol{e}_{mi} , \tag{11}$$

with the assumption that

$$\boldsymbol{e}_{mi} = \boldsymbol{Z}_i \boldsymbol{d}_i + \mathsf{e}_{mi} \sim \mathcal{N} \left( \boldsymbol{0}, \Sigma_{mi} \right) . \tag{12}$$

For the entire study population,

$$\boldsymbol{y}_m = \boldsymbol{X}_m \beta_m + \boldsymbol{e}_m . \tag{13}$$

With the assumptions that all independent sampling units have the same number of outcomes, i.e., $p_i = p$, and that $\Sigma_{mi} = \Sigma_M$ for all $i$,

$$\boldsymbol{e}_m \sim \mathcal{N}_{N \cdot p} \left( \boldsymbol{0}, \boldsymbol{I}_N \otimes \Sigma_M \right) . \tag{14}$$

*General linear mixed model hypothesis.* A general linear mixed model hypothesis about fixed effects can be written in terms of $\theta_m = C_m \beta_m$ as

$$H_{0m} : \theta_m = \theta_{0m} . \tag{15}$$

*Transforming from the multivariate model to the population-average reversible mixed model.* For all of the data, the multivariate model can be transformed to a reversible mixed model using the following equations:

$$y_m = \text{vec}\left(Y'\right) , \tag{16}$$

$$X_m = X_M \otimes I_p , \tag{17}$$

and

$$e_m = \text{vec}\left(E'\right) . \tag{18}$$

The hypotheses are also equivalent, as shown by the following equations:

$$\beta_m = \text{vec}\left(B'_M\right) , \tag{19}$$
$$C_m = \left(C_M \otimes U'\right) , \tag{20}$$
$$\theta_m = \text{vec}\left(\Theta'_M\right) , \tag{21}$$

and

$$\theta_{0m} = \text{vec}\left(\Theta'_{0M}\right) . \tag{22}$$

## 3 Updates

### 3.1 Algorithm updates

*How to use GLIMMPSE.* Users can access GLIMMPSE with a standard web browser via the url:

https://glimmpse.samplesizeshop.org/

Users are asked to complete a login, to allow tracking of the software use for future grant submissions.

GLIMMPSE may be redistributed or modified under the terms of the GNU General Public License version 2 (Free Software Foundation 2010). The source code for our Angular and Flask web applications and our functional statistical library which it relies upon are available for download from our GitHub Repositories, as is our collection of statistical functions packaged in the library pyglimmpse. The collection of statistical functions is also available at https://pypi.org/project/pyglimmpse/. Our GitHub Repositories have the following urls:

https://github.com/SampleSizeShop/glimmpseV3Front
https://github.com/SampleSizeShop/glimmpseV3Back
https://github.com/SampleSizeShop/pyGlimmpse .

Angular is a free and open-source framework for single-page web applications. One important reason for switching to Angular was to insure compatibility with future versions of web

browsers. Angular supports versions of Chrome, Firefox, Edge, Safari, iOS, and Android (https://angular.io/guide/browser-support).

*An updated covariance model for designs with nested clusters.* GLIMMPSE previously used the covariance model for nested designs with clusters given in Eq 7 of Chi et al. [10]. However, the Chi et al. [10] covariance is not the commonly used covariance for designs with clusters. In GLIMMPSE Version 3, an extension of Longford [12] variance components model is used instead.

In Eq 2.4, Longford [12] discussed a variance components model for a random effect design with every level containing only an intercept term. Longford [12] refers to this approach as a mixed analysis of variance model.

GLIMMPSE uses an extension of the [12] model. With $P_0 = 1$, $D$ nested levels of clustering, $k_d$ observations at level $d$ and $k_{*(D)} = \prod_{d=1}^{D} k_d$ total observations, a recursive computational algorithm for the Longford [12] correlation matrix for the responses from a single independent sampling unit is given by

$$P_{(D)} = I_{k_D} \otimes P_{(D-1)} + \rho_D \left( \mathbf{1}_{k_*(D)} \mathbf{1}'_{k_*(D)} - I_{k_*(D)} \right) , \qquad (23)$$

with corresponding covariance given by $\Sigma_D = \sigma^2 P_{(D)}$.

Eq 23 leads to an expression for the variance of the average of all observations in cluster $i$, which extends the Spearman-Brown prophecy formula [41,42]. With $k_{(D)} = [k_1 \cdots k_D]'$ and $\rho_{(D)} = [\rho_1 \cdots \rho_D]'$, and writing $k_{*(D)} = k_D \cdot k_{*(D-1)}$ and $k_{*(D)}^{-1} \mathbf{1}_{k_*(D)} = k_D^{-1} \mathbf{1}_{k_D} \otimes k_{*(D-1)}^{-1} \mathbf{1}_{k_*(D-1)}$, the variance of the average of all observations in cluster $i$ is

$$\sigma_{\bar{y}_i}^2 = \sigma^2 \cdot \gamma \left[ k_{(D)}, \rho_{(D)} \right] , \qquad (24)$$

with

$$\gamma \left[ k_{(D)}, \rho_{(D)} \right] = \gamma \left[ k_{(D-1)}, \rho_{(D-1)} \right] k_D^{-1} + \rho_D \left( 1 - k_{*(D)}^{-1} \right) , \qquad (25)$$

under the assumption that $\gamma \left[ k_{(0)}, \rho_{(0)} \right] = 1$. The expression equals the variance factor associated with the Spearman-Brown prophecy formula. For $D = 1$, the expression equals the "design effect," divided by the cluster size.

*An efficient algorithm for large clusters.* GLIMMPSE allows users to input designs with multivariate outcomes, repeated measures, and clustering. The covariance used assumes a direct product relationship, with the full variance, $\Sigma_y$, written as the Kronecker product of the covariance of the multivariate outcomes, $\Sigma_o$, the covariance of the repeated measures, $\Sigma_r$, and the clusters, $\Sigma_c$, so that

$$\Sigma_y = \Sigma_o \otimes \Sigma_r \otimes \Sigma_c . \qquad (26)$$

The Kronecker product of the corresponding within independent sampling unit contrasts for the outcomes, $U_o$, the repeated measures, $U_r$, and the clusters, $U_c$, yields the interaction contrast

$$U_y = U_o \otimes U_r \otimes U_c . \qquad (27)$$

The covariance of the outcomes, $\Sigma_y$, can become very large, very quickly. For example, with 8 and 10 repeated measures of each outcome, the resulting covariance matrix is of dimension $80 \times 80$. Adding clustering can further increase the size of the covariance matrix. The power and sample size calculations require computing the matrix product $\Sigma_* = U'_y \Sigma_y$. The

fastest matrix multiplication algorithms, are $\mathcal{O}(2.376)$, a speed not usually attained in Python. Instead, GLIMMPSE Version 3 computes

$$\Sigma_* = \boldsymbol{U}_o'\Sigma_o\boldsymbol{U}_o \otimes \boldsymbol{U}_r'\Sigma_r\boldsymbol{U}_r \otimes \boldsymbol{U}_c'\Sigma_c\boldsymbol{U}_c , \tag{28}$$

which typically has much smaller dimensions in each computation. For the example described above, with the multivariate analysis of variance hypothesis, randomization of the independent sampling units, and testing at time 3, $\boldsymbol{U}_o'\Sigma_o\boldsymbol{U}_o$ is $8 \times 8$, while $\boldsymbol{U}_r'\Sigma_r\boldsymbol{U}_r$ and $\boldsymbol{U}_c'\Sigma_c\boldsymbol{U}_c$ are both $1 \times 1$, with $\Sigma_*$ of dimension $8 \times 8$. The sizes of the product matrices are invariant to the cluster sizes. This means that users can conduct power and sample size calculations with essentially arbitrary cluster sizes. The current version sets a limit of $10,000$ elements per level of clustering. As a test, a run of the example in subsection 4.5 with $10,000$ schools per neighborhood ran just as fast as with 3 schools per neighborhood, in less than 3 seconds clock time.

## 3.2 Software updates

*Architecture updates.* In tandem with the new features, the interface and low level libraries have been rebuilt. GLIMMPSE Version 3 is a single page web application. The front end was written in Angular, with a Python Flask back end. The architecture is shown in Fig 1. Containerization with Docker and Docker Compose were used to allow portability. User logins, constructed using Auth0, permit tracking of use.

*Interface updates.* The interface was completely re-designed using a consistent palette and design. As an overall design choice, each screen was simplified. Two user studies, with 10 participants per study, allowed finding errors and optimizing the interface. The studies were considered to be exempt, quality-improvement projects by the University of Florida Institutional Review Board.

In the revised interface, each screen has a separate URL, which allows identification and tracking of bugs. A novel graphic design shows users the results of their design choices when they specify designs with multiple levels of clustering.

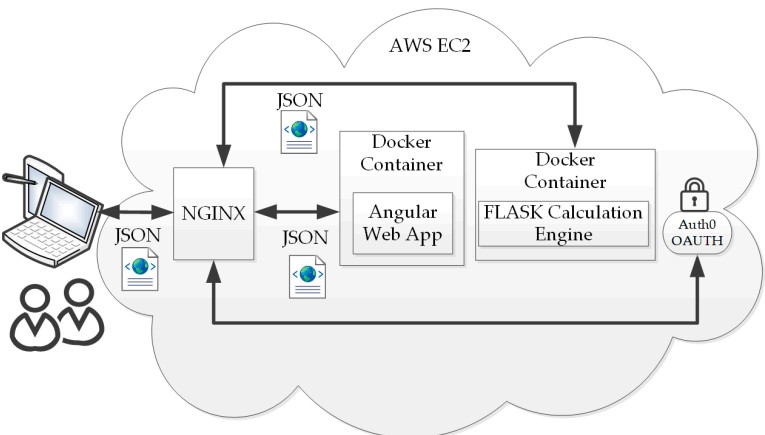

**Fig 1. Overview of the GLIMMPSE Version 3 architecture.**

User test feedback on the user interface and user experience (UI/UX) showed that users were confused by how to enter levels of clustering. In response, we added the graphical feedback shown in Fig 2. The figure shows an example design with children nested within classrooms, classrooms nested within grades, and grades nested within school. The element on the left shows the highest level of clustering. For this example, it is school. Moving to the right in the schematic shows lower levels of clustering. The schematic is telegraphic. It leaves out some of the classrooms within schools, and some of the children within classrooms. By omitting some information, the schematic shows the clustering clearly, but still fits on the screen. The schematic design was developed and finalized after multiple iterations of exchange between the software designers and the users.

A similar user-focused design approach was used to generate the screen allowing the user to select between hypotheses. The interface permits choosing within independent sampling unit hypotheses, between independent sampling unit hypotheses, or interaction hypotheses, as shown in Fig 3.

Most of the structure of the design is specified before the hypothesis screen appears. In order, GLIMMPSE prompts users to specify the following features of the study design.

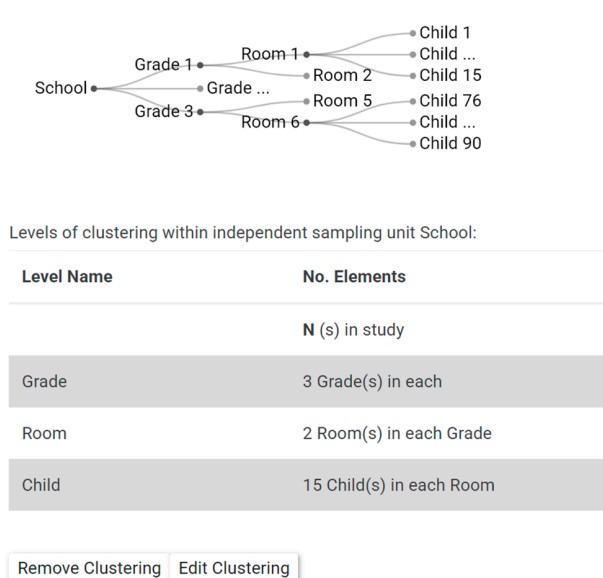

Levels of clustering within independent sampling unit School:

| Level Name | No. Elements |
|---|---|
| | **N** (s) in study |
| Grade | 3 Grade(s) in each |
| Room | 2 Room(s) in each Grade |
| Child | 15 Child(s) in each Room |

Remove Clustering    Edit Clustering

**Fig 2. Cluster input screen example.**

Select a hypothesis from the list.

| | Effects Available for Consideration | Nature of Variation |
|---|---|---|
| ◉ | Treatment x Time: Interaction | Between x Within |
| ○ | Time: Main Effect | Within |
| ○ | Treatment: Main Effect | Between |
| ○ | Grand Mean | Between |

**Fig 3. Hypothesis screen, showing between, within, and between-by-within hypotheses.**

- A study title.
- A choice between power and sample size.
- The test statistic.
- The Type I error rate.
- The study outcome or outcomes.
- The presence and nature of repeated measures of the outcome or outcomes.
- The presence and nature of clustering.
- The presence and nature of any fixed predictors.
- Whether there is a Gaussian covariate.

These features of study design permit the software to identify what kinds of hypotheses are possible. All possible hypotheses are listed on the screen. Radio boxes permit the users to choose the hypothesis for which they are computing power or sample size. The utility and clarity of this screen were enhanced by repeated interactions between the software team and participants in the user studies.

## 4 Examples

### 4.1 Overview

We provide five examples using GLIMMPSE Version 3. The examples are of increasing complexity, starting with a cluster randomized trial, and finishing with a subgroup analysis in a study with both clustering and longitudinality. The examples were developed and revised while teaching a short course on power and sample size for multilevel and longitudinal models. For didactic reasons, we chose fictional examples. The goal was to remove the complications which always occur in doing power analysis in actual study designs. The inputs chosen for the examples, including means, variances, standard deviations, sample sizes, powers, Type I error rates, correlations, covariates, and correlations were selected for clarity. They are not intended to be the correct parameters for the distributions of real variables.

The five examples were developed for students with a wide range of statistical abilities. Experienced statisticians and software engineers may find the descriptions too simplistic, and may prefer to simply use the .json files as a reference.

For each example, we provide a short vignette of the study, including the proposed hypothesis. We also provide a brief statistical analysis plan and a power or sample size analysis. The replication materials contain .json files which produce the power calculations in the examples.

### 4.2 Example 1: Power for a single-level cluster-randomized trial

*Study vignette.* A cluster randomized study was planned to examine the efficacy of a workplace training program to reduce alcohol consumption. Researchers planned to randomize workplaces to one of two treatment groups. During the study, the entire workplace would receive the same treatment. The goal of the study is to compare a workplace training program to a control treatment, in which there will be no training at all in the workplace. The outcome measure is the average number of drinks per day after the intervention. A flowchart for the study is shown in Fig 4. Fig 4 shows a simplified schematic of the study, using standardized flow chart symbols. After study start, workplaces are randomized to workplace training, or no training, and then drinking is evaluated.

The independent sampling unit is the workplace. Within each workplace, the responses of the workers are correlated. This occurs because the workers talk with each other in the workplace, may choose to drink together, may compare drinking activities, and will either undergo

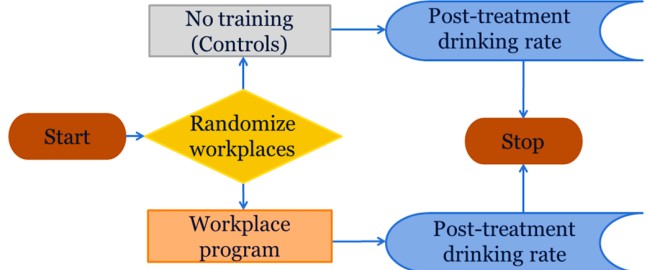

**Fig 4. A cluster-randomized controlled clinical trial of a workplace training program, with the average number of drinks per day as the outcome.**

workplace training, or no training, together. The unit of randomization is the workplace. The unit of observation is the drinking rate for each worker.

It is expected that the average numbers of drinks per day for the different workers in the same workplace are correlated. Thus if participants Able and Baker both work at University Hospital, it is expected that their average number of drinks per day are correlated. We assume that the different workplaces are independent. If participant Charles works at Children's Hospital, the average number of drinks per day for participant Charles should be independent of that of the average number of drinks per day of participants Able and Baker.

The between independent sampling unit factor is treatment. The between independent sampling unit factor has two levels: a workplace training program and a control program.

For the proposed study, every workplace is the same size and has 15 workers. No other covariates are measured. There will be 20 workplaces assigned to each treatment program, for a total of 40 workplaces. From previous clinical experience, it is speculated that none of the workplaces will drop out of the study. In addition, previous experience suggests that none of the workers will drop out of the study. Thus, post-treatment drinking rate will be measured on 600 people. Here, the number 600 is obtained by conducting the following calculation:

$$2 \text{ treatments} \times 20 \text{ workplaces per treatment} \times 15 \text{ workers per workplace} = 600 \text{ workers.} \quad (29)$$

From knowledge about the efficacy of the workplace intervention, the researchers speculate that the average number of drinks per day for the control workplaces will be 1.24. The average number of drinks per day for the workplaces where workers received treatment will be 0.73. The difference between 0.73 and 1.24 is considered to be of scientific interest. The common standard deviation of the measurement for each worker is expected to be 1.1.

The intracluster correlation coefficient will be 0.13. The intracluster correlation coefficient is a number between −1 and 1 which represents the correlation between the post-treatment drinking rate of two workers within one cluster.

*Hypothesis.* The null hypothesis is that there will be no difference in average number of drinks per day after intervention between workers who received no training and workers who received the workplace program. The alternative hypothesis is that the training program will change the average number of drinks per day. The researchers hope that the workplace training program will reduce the average number of drinks per day, making the average number of drinks per day smaller post-intervention in the treatment group relative to the control group. However, they would like to test for both larger and smaller post-intervention average

number of drinks per day. The researchers thought that there would be no change in average number of drinks per day at all in the control group.

*Analysis plan.* The analysis will use a general linear mixed model. The outcome variable will be the post-treatment drinking rate. As predictors, the model will include two indicator variables. The first indicator is one if there is a workplace training program, and zero if there is no workplace training program, and the workplace is in the control group. The second indicator is one if the workplace is in the control group, and zero if it is in the workplace training program. To account for correlation within the workplace, the model will include a random effect for workplace. This produces a compound symmetric error variance matrix. A Wald test with Kenward and Roger [39] degrees of freedom will be used, at a Type I error rate of 0.05.

*Power analysis.* For a general linear mixed model, with the drinking rates for each workplace as the outcome, and indicator variables for control group or workplace treatment program as predictors, and a Type I error rate of 0.05, the power for the Wald test will be 0.909.

## 4.3 Example 2: Sample size analysis for a longitudinal randomized trial

*Study vignette.* Researchers plan to conduct a longitudinal randomized controlled clinical trial in patients who had experienced a root canal. The outcome of interest is the memory of pain. The goal of the study is to determine if dental patients who were instructed to use a sensory focus have a different pattern of long-term memory of pain than participants who did not. Researchers hypothesize that the pattern of memory of pain would be different for those who had the intervention, and those who were in the control group.

Participants are to be selected and randomly assigned to either the sensory focus intervention or the standard-of-care intervention. An equal number of patients will be assigned to each treatment group. Patients in the intervention group will listen to automated audio instructions to pay close attention only to the physical sensations in their mouth. Patients in the standard-of-care group will listen to automated audio instructions on a neutral topic to control for media and attention effects.

All patients will be queried three times about their memory of pain. They will be asked to describe their memory of pain immediately, at six months, and at twelve months after the root canal and intervention. A flow chart for the study is shown in Fig 5. The flowchart provides a simplified schematic of the randomized trial. It shows study initiation, randomization to two groups (intervention and standard-of-care), and three longitudinal repeated measures on each study participant, before study end.

In this study, the outcome measure is the memory of pain. The independent sampling unit is the patient. The unit of randomization is the patient. The unit of observation is the memory of pain at each time point. It is expected that the three longitudinal measures over time for each patient will be correlated. It is also expected that each study participant will be independent from other study participants. The between independent sampling unit factor is treatment. Treatment has two levels: sensory focus intervention and control treatment. The within independent sampling unit factor is time. Time has three levels: zero months, six months and 12 months. It is expected that repeated measurements within each person will be correlated.

From previous research, the investigators thought that the memory of pain intensity at one week and 18 months had a correlation of 0.4. Given the previous research, for this example, it is assumed that the correlation between measures six months apart will be 0.5. Again from previous research, it is assumed that the correlation between measures 12 months apart will be 0.4. Pain assessed on a five point scale is assumed to have a standard deviation of 0.9. Based on clinical expertise, the investigators speculate that the pattern of means for the two groups will be as shown in Table 2.

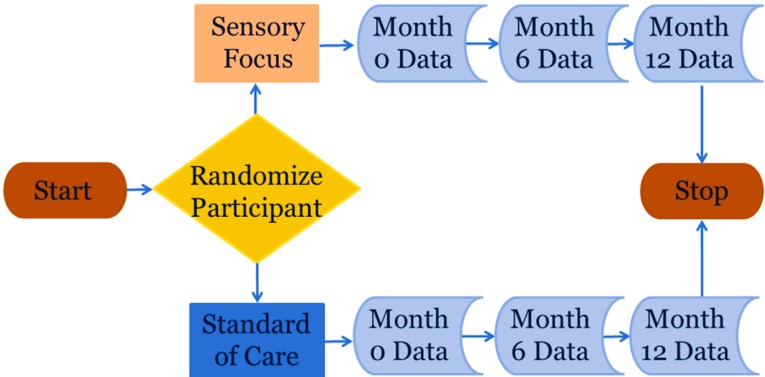

**Fig 5. A longitudinal randomized controlled clinical trial of a sensory focus intervention, with the memory of pain as the outcome.**

**Table 2. Predicted mean outcome for memory of pain score by treatment and time.**

|                  | Baseline | 6 months | 12 months |
|------------------|----------|----------|-----------|
| Sensory Focus    | 3.6      | 2.8      | 0.9       |
| Standard of Care | 4.5      | 4.3      | 3.0       |

The goal is to calculate the power for a sample size of 38.

*Hypothesis.* The null hypothesis is that the pattern of memory of pain over time would be no different between those who had the intervention, and those who were in the control group. The alternative hypothesis is that the pattern of memory of pain over time would be different for the control group and the intervention group. This is an interaction hypothesis, also known as a between-by-within hypothesis. An illustration of a time-by-intervention interaction is shown in Fig 6. A hypothesis about a time-by-intervention interaction asks the question, "Does the pattern of response in the outcome to treatment differ over time?" The figure shows a difference in patterns for the two randomization groups over time. Rejecting the null hypothesis of no time-by-intervention interaction corresponds to concluding that the same groups do not significantly differ in their pattern of outcomes over time.

*Analysis plan.* The investigators plan to fit a general linear mixed model. The outcome variables will be the three repeated measurements of memory of pain. The predictors will be two indicator variables, which, respectively, take on the value one if the person was assigned to sensory focus, and zero otherwise, and take on the value one if the person was assigned to standard-of-care, and zero otherwise. A Wald statistic with Kenward and Roger [39] degrees of freedom will assess the null hypothesis that the pattern of memory of pain over time is no different between those who had the intervention, and those who were in the control group. An unstructured covariance matrix and a Type I error rate of 0.05 will be assumed.

*Power analysis.* For a sample size of 38, the computed power is about 0.906, with the input values, model, and analytic plan shown, and a Type I error rate of 0.05.

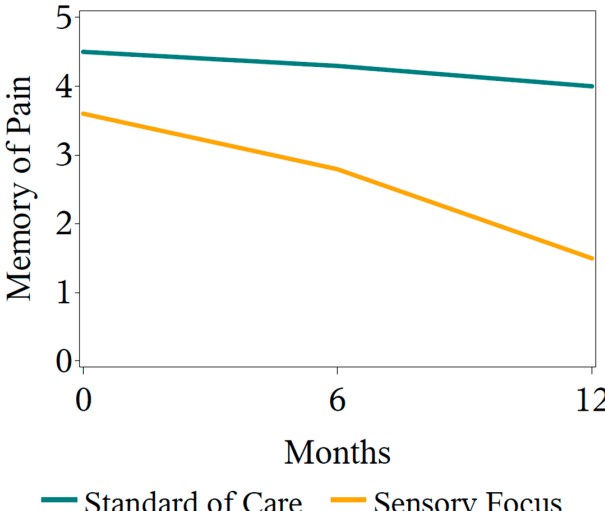

**Fig 6. A graph of the possible outcomes over time for the memory of pain trial.** The pattern of outcomes over time differs between the two randomization groups, a pattern consistent with time-by-intervention interaction.

### 4.4 Example 3: Sample size for a randomized trial with students nested within classrooms, and classrooms nested within schools, and a multivariate outcome

*Study vignette.* Researchers plan to conduct a randomized controlled clinical trial of an intervention designed to help young children learn fundamental early literacy skills. Alternative approaches include two standardized training options for teaching literacy, a monolingual approach and a bilingual approach.

Researchers plan to randomize 45 schools, with 15 in each treatment arm. That is, 15 schools will get the intervention, 15 schools will get a monolingual approach, and 15 schools will get a bilingual approach. All of the schools are very far apart from each other. In fact, they are so far apart that the students, parents and teachers in each school have no contact with the students, parents and teachers in any other schools. Thus, schools can be considered to be independent.

Initial study planning will begin by assuming each school has exactly four kindergarten classrooms, each with five students, all of whom will take part in the trial. Thus, each school has 20 total students, with (five students per classroom) × (four classrooms per school). In most real scenarios, there will be a different number of students in each classroom, a different number of classrooms in each school, and a different number of schools in each neighborhood.

The extent that gender, age and ethnicity of the students is related to the outcomes is not going to be considered for the initial study planning. The intraclass correlation coefficient for classrooms within schools was assumed to be 0.07. The intraclass correlation coefficient for students within classrooms was assumed to be 0.04.

Students will be evaluated on three composite outcome scores that are considered to have a multivariate normal distribution. Here, we designate the scores as Score A, Score B and Score C.

Students will be measured before and after the intervention, and for each component, a difference score will be calculated as the post-score, subtracted from the pre-score. The outcomes of interest are the difference scores for the three composite scores. Each student will contribute three difference scores: one for Score A, one for Score B, and one for Score C.

Initial study planning will begin by assuming no missing data, which corresponds to requiring everyone present for either the pre- or post-test is present for both. Subsequent refinement of the sample size analysis could include an allowance for missing data, if it is deemed appropriate. The scientists wonder what their power will be for the proposed trial, if the means and standard deviations are as shown in Table 3.

Based on knowledge of previous studies, scientists have a good guess as to what the correlation matrix looks like for the difference scores of the three outcome measures. The expected correlation matrix is shown in Table 4.

The common standard deviation (across the three intervention arms) for Score A is 4.4, for Score B is 4.2, and for Score C is 0.6.

*Hypothesis.* The three difference variables define a multivariate response profile. Scientists hypothesize that the three literacy training programs do not differ in any combination of the outcome differences.

*Analysis plan.* Scientists plan to fit a general linear mixed model with the 20 difference scores in Scores A, B and C respectively, observed per school as outcomes. There will be 45 schools contributing data to the model, with 15 assigned to each of three treatment arms. As fixed effect predictors, the scientists plan to use indicator variables for the three treatments. The covariance will be a Kronecker product of an unstructured covariance for the three difference scores, and a compound symmetric matrix for repeated values within the schools. The scientists plan to test a multivariate analysis of variance hypothesis. A Wald test with Kenward and Roger [39] degrees of freedom will be used, at a Type I error rate of 0.05, to evaluate the null hypothesis of no differences in response difference profiles among the three literacy programs.

*Power analysis.* Power will be 0.659 to evaluate the null hypothesis of no profile differences between the literacy programs, using a multivariate analysis of variance, and the Wald test with [39] degrees of freedom, at a Type I error rate of 0.05.

## 4.5 Example 4: Sample size for a multilevel trial with longitudinal repeated measures

*Study vignette.* Scientists want to find a sample size for a planned randomized controlled clinical trial. They are interested in power of at least 0.90. Researchers plan to conduct a

**Table 3. Predicted mean difference scores for three literary scale scores, stratified by intervention arm.**

|              | Score A | Score B | Score C |
|--------------|---------|---------|---------|
| Intervention | 0.3     | 0.3     | 0.3     |
| Monolingual  | 0.1     | 0.1     | 0.1     |
| Bilingual    | 0.1     | 0.1     | 0.1     |

**Table 4. Correlation matrix for difference scores.**

|         | Score A | Score B | Score C |
|---------|---------|---------|---------|
| Score A | 1.0     | 0.9     | 0.2     |
| Score B | 0.9     | 1.0     | 0.4     |
| Score C | 0.2     | 0.4     | 1.0     |

randomized controlled clinical trial of an intervention designed to reduce adolescent alcohol use. The goal was to compare the intervention with no intervention. After obtaining consent from students, parents, teachers, and administrative staffs, researchers grouped schools within neighborhoods to form neighborhood groups. Neighborhood groups were randomized to either the alcohol use intervention or standard of care, using a two to one randomization scheme. The two-to-one randomization scheme has two times the number of neighborhoods randomized to the workplace training program, as a result of community demand. For logistic and cost reasons, as well as the desire to ensure the smallest group is of sufficient size, the scientists wish to restrict total sample size to the range of 30–45 neighborhoods.

Each student will be surveyed at baseline and again three more times after the treatment begins. Thus there will be four measurements, at baseline, in the spring of sixth grade, the spring of seventh grade, and the spring of eighth grade. The survey will use a detailed alcohol intake recall method to obtain an alcohol use scale for each student at each time point. Previous similar modeling with this variable has produced acceptably normally distributed jackknifed studentized residuals without transformation of the data.

For the purposes of this question, we can assume that there were an equal number of students in each classroom, an equal number of classrooms per school, and equal number of schools per neighborhood group. We expect all neighborhoods, schools and classrooms to stay with the study throughout the entire time. However, we know that students move in and out of the district. Previous studies of student absences has reassured us that the absences of the students are not related to the neighborhood, classroom or school, nor to the use or non-use of the alcohol treatment program, nor to the age of the student.

There are 20 students in each classroom, with intracluster correlation of 0.09. There are three classrooms per school, with an intracluster correlation coefficient of 0.04. There are two schools per neighborhood, with an intracluster correlation coefficient of 0.03.

The researchers believe, from previous experience with similar data, that the correlation of scores across time followed a LEAR model with base correlation of 0.6 and a decay rate of 0.7, leading to a decrease in correlation of about 0.10 per unit time. The LEAR model allows correlation to be strong at first, and then die off with time at a rate controlled by the decay parameter [43]. The AR(1) model is a special case. Measurements will be conducted at zero, one, three and five months.

From previous experience and work, the researchers are interested in the pattern of means shown in Table 5, with common standard deviation of 4.

*Analysis plan.* Scientists plan to fit a general linear mixed model with the alcohol use scores for each student as the outcomes. As predictors, they will use indicator variables for the two treatments, the workplace training program and the standard of care. The scientists plan to account for correlation of schools within neighborhood groups, classrooms within schools, and students within classrooms. In all three levels, the schools are assumed to exchangeable within neighborhoods, the classrooms within schools, and the students within classrooms. This assumption leading to compound symmetry for each level of clustering. The longitudinal repeated measures of alcohol use over time will assume a LEAR covariance structure [43].

**Table 5. Predicted neighborhood mean alcohol use scores, by intervention arm.**

|  | Month 0 | Month 1 | Month 3 | Month 5 |
|---|---|---|---|---|
| Workplace Training Program | 5.2 | 5.3 | 5.3 | 5.3 |
| Standard of Care | 5.2 | 5.5 | 5.9 | 6.2 |

Scientists plan to use a Wald statistic with [39] degrees of freedom and a Type I error rate of 0.05 to evaluate the null hypothesis of no difference in pattern of average alcohol use scores over time between the treatments.

*Power analysis.* The two-to-one sampling ratio for the treatment to the comparison group makes the only possible sample sizes divisible by three. For a sample size of 102, the power is about 0.909.

## 4.6 Example 5: Power analysis for a planned subgroup analysis

*Study vignette.* Scientists plan to measure an imaging outcome in two regions of the brain. It is expected that the response to an intervention or control will differ by genotype at a specific single nucleotide polymorphism. Here, we code the genotypes as A, B, C and D. The expected means and standard deviations are shown in Table 6.

For a conservative power analysis, the estimate of the standard deviation used is rounded up to 0.3. The power analysis uses the inputs shown in Table 7, with 'Left' indicating the mean left middle cerebral imaging measure, and 'Right' indicating the mean right middle cerebral imaging measure.

The correlation between summary imaging measures for the two brain regions are assumed to be as shown in Table 8.

*Hypothesis.* The null hypothesis is no genotype-by-treatment interaction on the imaging measurements for the two brain regions.

*Power analysis.* With a total sample size of 30, the power is about 0.792.

# 5 Accuracy and timing of the power calculations

## 5.1 Simulation methods to assess accuracy

The goal was to assess the accuracy of GLIMMPSE Version 3 against Monte Carlo simulations. We used the five examples included in Sect 4. For each example, we simulated data from a general linear multivariate model, defining the predictor matrix, the within independent sampling unit contrast, the between independent sampling unit contrast, the matrix of slopes and intercepts, and the covariance of the errors. In all examples considered, the null hypothesis was that linear combinations of the parameter were equal to zero. We fixed $\alpha$. We simulated a random error matrix, and formed $\mathbf{Y_M}$ using Eq 1.

**Table 6. Statistics for summary imaging measures, by brain region.**

|  | Mean | Standard Deviation |
|---|---|---|
| Left Middle Cerebral Group | 3.12 | 0.232 |
| Right Middle Cerebral Group | 3.17 | 0.269 |

**Table 7. Predicted responses by intervention and genotype.**

| Treatment | Genotype | Left | Right |
|---|---|---|---|
| Placebo | A | 3.12 | 3.17 |
| Placebo | B | 3.12 | 3.17 |
| Placebo | C | 3.12 | 3.17 |
| Placebo | D | 3.12 | 3.17 |
| Chemo | A | 2.10 | 2.15 |
| Chemo | B | 2.30 | 2.35 |
| Chemo | C | 2.50 | 2.55 |
| Chemo | D | 2.70 | 3.25 |

**Table 8. Correlation between summary imaging measures for different brain regions.**

| Region | Left | Right |
|---|---|---|
| Left | 1.00 | 0.53 |
| Right | 0.53 | 1.00 |

We computed estimates, and conducted hypothesis testing using the Hotelling-Lawley Trace statistic. We compared the observed test statistic to the McKeon reference distribution to obtain a $p$ value. The process was repeated for $10,000$ replications. Empirical power was computed as the proportion of p-values smaller than the stipulated $\alpha$ level, i.e., the proportion of rejections of the null hypothesis. With $10,000$ replications, the half width of the 95% confidence interval for the empirical power estimate, $\hat{p}$, was no more than $1.96 \times \sqrt{\hat{p}\left(1 - \hat{p}\right)/10000}$. Here 1.96 is the 97.5% quantile from a normal distribution. The error in the estimation of power was, at most, at the second decimal, with all computed values falling within the estimated 95% confidence intervals.

The replication materials use backend calculations. The code package for the replication is available at https://github.com/SampleSizeShop/glimmpseV3back. For user convenience, in Supplementary Material A, we have also included .json files and screenshots of the results, for the computation of the power values with the online version of GLIMMPSE.

## 5.2 Simulation methods to assess timing

Total CPU time was assessed for GLIMMPSE calculations and Monte Carlo simulations using timing in the Python code. The timing values reflect calculations and simulations performed on a 2017 MacBook Pro, with 3.1 GHz Quad-Core Intel Core i7, and 16GB RAM, under the operating system MacOS Catalina Version 10.15.7. Timing was assessed on power calculations which were computed using the Python libraries, without the web interface. The timing values thus do not include time for HTTP processing by the web interface.

## 5.3 Accuracy and timing results

Accuracy and timing results for the five examples appear in Table 9, where L95 and U95 are the lower and upper bounds, respectively, of the 95% confidence intervals.

The maximum absolute deviation from the Monte Carlo simulation was no more than 0.01. Timing results for calculations and simulations appear in units of seconds. Timing results for simulations are on the order of about $1,000,000$ times slower.

**Table 9. Accuracy and timing, compared to simulation. Time is in seconds.**

| Example # | GLIMMPSE Time | Sim. Time | GLIMMPSE Backend | Mean of Sim. | L95 | U95 |
|---|---|---|---|---|---|---|
| 1 | 0.0012 | 2662 | 0.909 | 0.910 | 0.903 | 0.915 |
| 2 | 0.0007 | 3466 | 0.906 | 0.902 | 0.900 | 0.911 |
| 3 | 0.0007 | 4039 | 0.659 | 0.657 | 0.649 | 0.668 |
| 4 | 0.0009 | 6388 | 0.909 | 0.905 | 0.904 | 0.915 |
| 5 | 0.0010 | 3289 | 0.792 | 0.796 | 0.784 | 0.799 |

# 6 Evaluating the impact of GLIMMPSE

## 6.1 Methods

*Choosing evaluation metrics.* Afiaz et al. [44] discussed best practices for evaluating the impact of biomedical software. Evaluation metrics for GLIMMPSE were chosen following the Afiaz et al. [44] suggestions. Some metrics, such as unique downloads, were not appropriate for a software used mainly through a web interface. Instead, of the metrics listed by Afiaz et al. [44], we chose to evaluate usage, user satisfaction, citations, and a measure of financial impact.

*Assessing usage.* GLIMMPSE has a series of pages. Each page has its own Uniform Resource Locator (URL). Visits to each page are tracked via Google Analytics (Version G4). Tracking began June 15, 2023.

In general, it is difficult to differentiate between true human users, and bots. For GLIMMPSE, usage counts likely reflect human users, for two reasons. First, users must create a unique login with Auth0, which must be validated via an email link before it becomes active. Second, each power or sample size calculation in GLIMMPSE requires multiple user interactions, on multiple screens. The calculate button is the final step before a user receives power and sample size results. The calculate button only becomes active when all information about a study has been entered correctly. Usage was evaluated by recording the number of times the calculate button was hit, over a year, from midnight, April 29, 2024 to April 29, 2025.

*Finding citers*: The total number of citers was computed on June 13, 2025. The first release of GLIMMPSE was described in two publications, Kreidler et al. [11] and Guo et al. [45]. Kreidler et al. [11] appears in a "How to cite button" at the top of the www.SampleSizeShop. org page. Guo et al. [45] is a tutorial which appeared in the journal *BMC Medical Research Methodology*. Citations were searched for using the "cited by" field in PubMed, using the Stats/Citations functions in ResearchGate, and in Google Scholar citations. In addition, searches were conducted using the GLIMMPSE Research Resource Identifier (RRID: SCR_016297). All citations and RRID mentions were downloaded as pdf files, to create a corpus of citing documents.

*Citer inclusions and exclusions.* The total financial impact was computed on June 13, 2025. Citations were only included in the count of citers and the calculations of financial impact if 1) the citations were peer reviewed and 2) the citations used GLIMMPSE for an *a priori* power or sample size calculation. Citers were excluded if they were published in the following formats: conference papers, books, masters papers, theses or dissertations. In addition, the calculations excluded methods papers, including tutorials, derivations, and examinations of accuracy. If the citation was used to justify the sample size, but no calculations appeared to have been done, the citers were excluded. When the decision was unclear, the citations (and their funding grants) were excluded. We excluded all U grants that provide foundational funding for Clinical and Translational Science Institutes (NCATS CTSI). Total citing documents considered, included, and excluded were tracked as suggested in the Strengthening the Reporting of Observational Studies in Epidemiology (STROBE) statement [46].

*Finding citers with United States National Institutes of Health funding.* To find those with United States NIH funding, we searched the corpus of citing documents using Agent Ransack Pro (Version 2019, Build 2938, Mythicsoft, Cambridge, United Kingdom). The Boolean search strategy looked at character strings which started within 100 characters of the words "Funding" or "Supported" or "Funded." The Boolean search looked for any listing of 1) Any United States federal Funding agency name or abbreviation AND 2) Any of the three letter alphanumeric grant codes used by federal funders since 2013 (e.g. R01, K01, or U01). The search strategy appears in Supplementary Material B: Search Strategy and Grant List.

*User survey reported satisfaction and funding.* On March 20, 2025, we sent out a survey to registered users, using the REDCAP electronic data capture tools hosted at the University of Colorado Denver [47]. The survey was reviewed by the Colorado Multiple Institutional Review Board (COMIRB number 25-0593) and deemed to be not human subjects research, because it involved a quality improvement survey which was not generalizable. The survey included a question on user satisfaction, and a question inviting respondents to report whether they had used GLIMMPSE to design a study funded by the NIH. If respondents indicated that they had used GLIMMPSE to design a study funded by NIH, they were invited to enter the grant number as free text.

*Finding total funding, overall, and by NIH institutes.* To generate the total amount of funding used by grants which cited GLIMMPSE for design, the grant numbers from citers who had used GLIMMPSE for design were copied into NIH Reporter, in the project number ID box, selecting both all active projects, and all fiscal years in the Fiscal Year box. The search and the search criteria were saved to document the search. Charts and totals were obtained from the Charts button, which computed total funding across fiscal years for all projects for each institute and center, and overall.

## 6.2 Results

Google Analytics shows 14, 313 power or sample size calculations were performed over a year, from midnight, April 29, 2024 to April 29, 2025.

Of the 2146 users who had logged in at least twice, there were 268 responses. This is a 12% response rate. Some 249 (93%) of the users indicated that they would recommend GLIMMPSE to others.

The software has been cited 497 times since release in 2013. A bibliography of citing manuscripts is shown in Supplementary Material C: Software Citers. Citations per year are shown in Fig 7. Citations appear to have grown roughly exponentially since 2013, excluding edge effects from incomplete years, and lagging reporting per year.

The STROBE diagram [46] for inclusion and exclusion of citers for the funding calculation is shown in Fig 8. Of the citers meeting inclusion and exclusion criteria, 41 (8%) included citations to NIH funded grants. An additional 15 users reported funding from 11 NIH grants

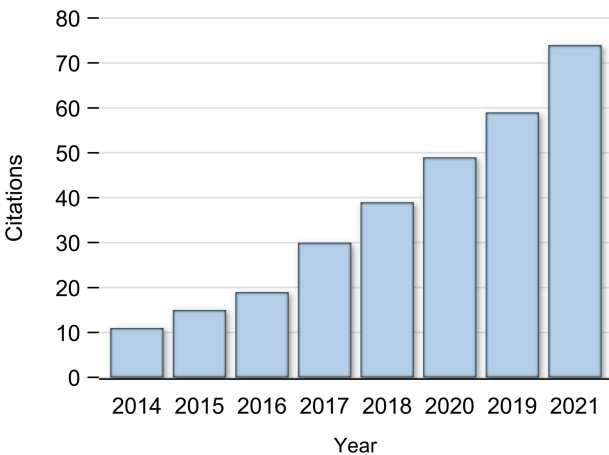

**Fig 7. Citations per year.**

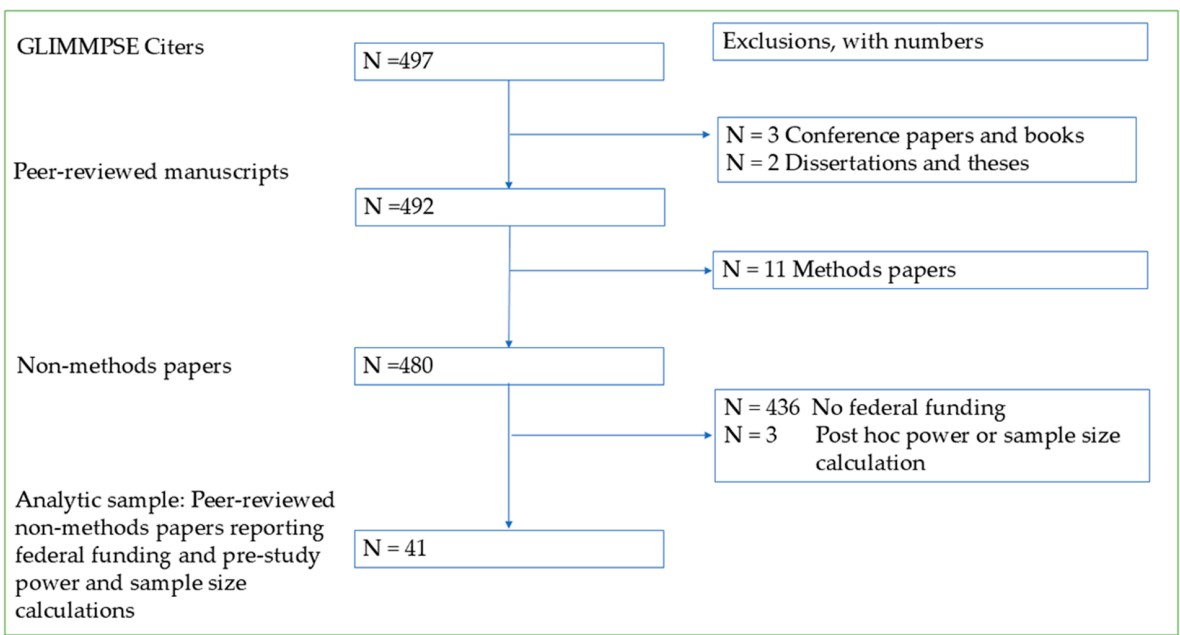

**Fig 8. Documents included in citation and financial calculations: STROBE[46] diagram.**

from our user survey. There was no overlap between the grants acknowledged in the publications of GLIMMPSE citers, and the grants reported in the user survey. All the grants reported are listed in Supplementary Material B: Search Strategy and Grants List. The grants were from 16 of the 27 NIH institutes.

Total funding for NIH grants designed using GLIMMPSE software by institute and center is detailed in Table 10. The total funding is now $229, 499, 156. The column on the left, labeled 'Institute,' indicates the administering institute or center.

## 7 Discussion

GLIMMPSE provides power and sample size calculations for a wide range of designs. While only randomized controlled trials are given in the examples, the package can also be used to compute power and sample size for observational studies, including those with longitudinal repeated measures, clustering, or both. The package can also be used for study designs beyond those described in the examples in the paper. GLIMMPSE will provide calculations for analysis of variance (ANOVA), analysis of covariance (ANCOVA), and studies with multiple repeated measures, among other designs. The package provides exact or accurate approximate calculations, using methods supported by many peer-reviewed publications (see, e.g., [5–7]). The ability to download power computations and save them for later work fosters reproducible research. Validation studies provide further support for the accuracy of the calculations. Because the software is free and open-source, and available on the web, an increasing number of publications have cited it. GLIMMPSE has long provided computations for heterogeneity of treatment effect, or heterogeneity of exposure-outcome associations, yet no previous publication has provided an example. This publication helps fill the gap.

GLIMMPSE has some limitations. GLIMMPSE applies only to data with Gaussian (normal) errors. It does not provide power and sample size computations for binary outcome data, nor for count data. We hope to add these features in the future. The use of the reversible

**Table 10. Total United States NIH funding of grants designed using GLIMMPSE software, by institute. The column labeled "Citers" shows funding acknowledged by authors who both cited GLIMMPSE and described a power or sample size calculation with GLIMMPSE in their manuscript. The column labeled "Survey" shows funding reported by those who responded to our user survey.**

| Institute | Citers | Survey | Total |
|---|---|---|---|
| NCATS | $21,943,019 | | $21,943,019 |
| NCI | | $3,950,143 | $3,950,143 |
| NHLBI | $44,697,080 | | $44,697,080 |
| NIA | $66,428,799 | | $66,428,799 |
| NIAAA | $4,771,191 | | $4,771,191 |
| NIAID | | $427,625 | $427,625 |
| NIAMS | $1,612,045 | | $1,612,045 |
| NICHD | $14,563,402 | $780,481 | $15,343,883 |
| NIDA | $1,963,069 | $3,083,363 | $5,046,432 |
| NIDCD | $441,025 | | $441,025 |
| NIDCR | $2,132,372 | | $2,132,372 |
| NIDDK | $6,744,602 | $1,541,664 | $8,286,266 |
| NIGMS | $3,574,099 | | $3,574,099 |
| NIMH | $41,445,245 | $4,664,297 | $46,109,542 |
| NINDS | $1,825,993 | | $1,825,993 |
| NINR | $2,909,642 | | $2,909,642 |
| **Total** | $215,051,583 | $14,447,573 | $229,499,156 |

model means that all outcomes, predictors and covariates are assumed to have complete data, and that all clusters are the same size. In practice, missing data occurs routinely. In the future, the hope is to incorporate into the software several power and sample size methods that can accommodate missing data, including those of Ringham et al. [36] and Kreidler et al. [40]. Ringham et al. [36] provided power calculations accounting for missing data for a modified test statistic for the multivariate model due to Catellier and Muller [48]. Kreidler et al. [40] gave power approximations for the Kenward and Roger [39] test for the mixed model, which can also allow for missing data. If the data can be assumed to be missing completely at random, scientists can account for missing data using a simple strategy. If the loss-to-followup rate is known to be $x$%, an adjusted sample size may be calculated by using the full data sample size, divided by the quantity $[1 - (x/100)]$. To examine the effect of unequal cluster sizes, scientists can compute bounds on power or sample size. If both the minimum and maximum expected cluster sizes are known, one can compute power and sample size under the assumptions of the minimum and maximum cluster sizes. The true power and sample size will be between the power and sample size estimates for the minimum and maximum cluster sizes.

Several new features expand capabilities. A complete, ontology-based reconfiguration of the interface was chosen to improve the human-computer interface. The ability to handle very large cluster sizes will improve the application to applications such as Medicaid data. The capability for separate between independent sampling unit and within independent sampling unit hypothesis specification improves clarity and utility.

Use statistics, citations, and United States National Institutes of Health (NIH) funding of studies citing GLIMMPSE continue to rise, showing robust demand. Power and sample size software will always show fewer citations than sites which aggregate routines for data analysis. Yet each citation of our software represents the design of an entire study, and thus represents the investment of great amounts of time and money. This is perhaps best illustrated by an example. One group citing our software, Boyne et al. [49], described a randomized controlled trial of a high-intensity interval training intervention in patients with chronic stroke

(NCT03760016). Boyne et al. [49] cited funding from R01HD093694, which came in at over 3.3 million United States dollars.

One measure of the utility of the GLIMMPSE software is the wide variety of fields of science in which our citers work. GLIMMPSE has been cited in fields ranging from cardiology to memory, from studies of sheep hematology to randomized trials in elder care. While we counted citations per year, and funding accrued by citers, the actual citations of GLIMMPSE citers give qualitative evidence of the wide impact on scientific research of the software program.

GLIMMPSE has been supported by multiple grants from the United States National Institutes of Health (NIH). We hope that such funding will continue. However, in the current funding environment, depending only on the NIH may not promote sufficient stability of the software. We have two plans for sustaining the software. First, we hope to leverage external funding to produce command-line callable, end-to-end Python libraries. We will encourage users to access the libraries at our GitHub repository. Second, we hope to explore opportunities to use a business approach to support software development. Setting up a certified B corporation, a societally-focused for-profit corporation, would permit paid sample size consulting to support the free software, under a "freemium model." We plan to survey GLIMMPSE users to examine the market size for such consulting. A stable funding source would permit frequent releases and updates, and standardized bug tracking.

No package can provide power and sample size calculations for every possible design. The new version of GLIMMPSE described here provides calculations only for designs with Gaussian errors, and even then, not all such designs. The software does not provide power and sample size calculations for cross-over designs or stepped wedge designs.

GLIMMPSE has four current gaps in capability, which we plan to target in the future. GLIMMPSE does not provide calculations for randomized complete block designs, nor any design with randomization within the cluster. GLIMMPSE does not provide calculations which allow users to compare the effects on power and sample size of choosing multivariate rather than composite outcomes. GLIMMPSE does not provide capability to support power calculations for high dimensional data, including microbiome, epigenetic, genetic, and metabolomic data. Finally, GLIMMPSE does not allow accounting for unequal cluster sizes. We hope to add these capabilities in the future.

## Supporting information

**S1 File. A. Includes json and screenshot files for the five examples described in the manuscript.**
(ZIP)

**S2 Text. B. Search And Grant List.**
(PDF)

**S3 Text. C. Software Citers.**
(PDF)

## Acknowledgments

*Support.* The use of REDCAP in this project was supported by NIH/NCATS Colorado CTSA Grant Number UL1TR002535. The content is solely the responsibility of the authors and does not necessarily represent the official views of the National Institutes of Health. The funders had no role in the study design, data collection and analysis, decision to publish, or preparation of the manuscript.

## Author contributions

**Conceptualization:** Deborah H. Glueck, Alasdair J. Macleod, Keith E. Muller.

**Data curation:** Deborah H. Glueck, Alasdair J. Macleod, Keith E. Muller.

**Formal analysis:** Deborah H. Glueck, Alasdair J. Macleod, Elizabeth M. Litkowski, Keith E. Muller.

**Funding acquisition:** Deborah H. Glueck, Keith E. Muller.

**Investigation:** Deborah H. Glueck, Alasdair J. Macleod, Keith E. Muller.

**Methodology:** Deborah H. Glueck, Alasdair J. Macleod, Elizabeth M. Litkowski, Keith E. Muller.

**Project administration:** Deborah H. Glueck, Alasdair J. Macleod, Elizabeth M. Litkowski, Jiang Bian, Albert D. Ritzhaupt, Keith E. Muller.

**Resources:** Deborah H. Glueck, Alasdair J. Macleod, Keith E. Muller.

**Software:** Deborah H. Glueck, Qian Li, Alasdair J. Macleod, Elizabeth M. Litkowski, Xi Yang, Jiang Bian, Keith E. Muller.

**Supervision:** Deborah H. Glueck, Alasdair J. Macleod, Albert D. Ritzhaupt, Keith E. Muller.

**Validation:** Deborah H. Glueck, Alasdair J. Macleod, Keith E. Muller.

**Visualization:** Deborah H. Glueck, Alasdair J. Macleod, Keith E. Muller.

**Writing – original draft:** Deborah H. Glueck, Alasdair J. Macleod, Keith E. Muller.

**Writing – review & editing:** Deborah H. Glueck, Qian Li, Alasdair J. Macleod, Elizabeth M. Litkowski, Jiang Bian, Albert D. Ritzhaupt, Max Sommer, Natercia Valle, Jessica R. Shaw, Keith E. Muller.

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
