## [Decision Letter · Decision Letter 0]

21 May 2025

PONE-D-24-57373Power and Sample Size for Balanced Linear Mixed

Models with Clustering and Longitudinality:

GLIMMPSE 3PLOS ONE

Dear Dr. Glueck,

Thank you for submitting your manuscript to PLOS ONE. After careful consideration, we feel that it has merit but does not fully meet PLOS ONE’s publication criteria as it currently stands. Therefore, we invite you to submit a revised version of the manuscript that addresses the points raised during the review process.

We look forward to receiving your revised manuscript.

Kind regards,

Abhik Ghosh

Academic Editor

PLOS ONE

Journal Requirements:

Reviewers' comments:

Reviewer's Responses to Questions

**Comments to the Author**

1. Is the manuscript technically sound, and do the data support the conclusions?

Reviewer #1: Yes

Reviewer #2: Yes

Reviewer #3: Yes

2. Has the statistical analysis been performed appropriately and rigorously? 

Reviewer #1: Yes

Reviewer #2: Yes

Reviewer #3: Yes

3. Have the authors made all data underlying the findings in their manuscript fully available?

Reviewer #1: Yes

Reviewer #2: Yes

Reviewer #3: Yes

4. Is the manuscript presented in an intelligible fashion and written in standard English?

Reviewer #1: Yes

Reviewer #2: Yes

Reviewer #3: Yes

5. Review Comments to the Author

Reviewer #1: Peer Review of

Power and Sample Size for Balanced Linear Mixed Models with Clustering and Longitudinality: GLIMMPSE 3

1. Summary of Manuscript

In this manuscript, the authors introduce an updated version of GLIMMPSE (now version 3), a free, web-based, open-source software tool designed to provide power and sample size calculations for studies that use Gaussian linear mixed models. The authors describe the theoretical foundations of the software (relying on established multivariate and mixed-model power analysis references) and highlight three significant new features:

1.Greatly Extended Nesting/Clustering Capability: Up to 10 levels of clustering, with an efficient recursive algorithm for larger cluster sizes.

2.User Interface Overhaul: A redesigned GUI, built in Angular with a Python Flask backend, employing a one-topic-per-screen design.

3.Updated Covariance Structures: Adoption of Longford’s (1987) variance components model for multi-level clustering and improved handling of between-within (or between-by-within) hypotheses.

The manuscript demonstrates these new features through five example studies, each increasing in complexity (from a basic cluster-randomized trial to a multi-level, longitudinal study). It concludes with the results of a Monte Carlo simulation study to assess accuracy and citation metrics showing the notable impact of GLIMMPSE in NIH-funded research.

This is an important methodological contribution, with robust theoretical grounding and practical tools to help researchers design studies requiring linear mixed-model analyses. Below, I provide an in-depth critique of the scientific content, methodology, and language.

2. Major Comments

1.Clarification of Reversibility Conditions

The manuscript emphasises the importance of “reversibility” in mixed models but could benefit from more explicit statements or examples of when these conditions are (and are not) met in practice. For instance, while the conditions are described mathematically, it would help readers to see a short paragraph explaining typical pitfalls—e.g., when cluster sizes differ or data are missing at some time points (which breaks the assumption of complete data).

oSuggested Revision: Add subsection in Section 2 (“The Reversible General Linear Mixed Model”) to give at least one real-world scenario in which reversibility fails and how that would impact power analysis.

2.Broadening Example Designs

The examples provided are highly relevant (and cover cluster-randomised, longitudinal, and multi-level designs). However, the text might explicitly mention other design types (e.g., crossover designs, stepped-wedge designs, or observational studies with repeated measures) and whether GLIMMPSE 3 can or cannot handle them.

oSuggested Revision: In Section 4 (“Examples”), add brief bullet points stating how the software might be used or not used for other popular designs (e.g., stepped-wedge trials).

3.Covariance Structures and Alternatives

The authors introduce a Longford-based variance components approach. While this is well justified, readers may wonder how robust the method is under certain misspecifications (e.g., if the true covariance is a partially autoregressive correlation of order 1, AR(1), or if only partial nesting is present).

oSuggested Revision: Add a short statement in Section 3.1 or Section 3.2, clarifying how sensitive the method is to mild covariance misspecification. Cite any existing references (e.g., Pinheiro & Bates, 2000; Laird & Ware, 1982; or relevant chapters from Linear Model Theory by Muller & Stewart, 2006).

4.Handling Missing Data

Although the authors note that the software handles designs under the assumption of complete data, many real-world studies encounter missingness. The authors mention that advanced methods exist (e.g., references to Josey et al., 2023), but the practical approach within GLIMMPSE 3 could be expanded.

oSuggested Revision: In the “Future Directions” or “Discussion” section, briefly note the limitations for missing data. Mention if future releases might incorporate approximations for missing data or reference existing guidance on incorporating dropout in sample size planning (e.g., by inflating sample size or adjusting design effect).

5.Software Usability and Maintenance

The authors highlight improvements to the interface and code refactoring in Python. A potential concern is ensuring long-term maintainability and stability (e.g., version control, bug tracking, frequency of releases).

oSuggested Revision: Add a brief paragraph in Section 3.2 clarifying the planned schedule for future updates or a statement about user/community involvement in the GitHub repository (e.g., encouraging pull requests).

3. Minor Comments and Suggested Edits

1.Language & Grammar

oSection 1.3 (“Literature Review”): Consider rephrasing “Here, we update the literature review...” to “Here, we update our previous review of relevant software...” for clarity.

oSection 4.1 (“Study Vignette”): The text “It is expected that the results for the different workplaces are independent” could be simplified to “We assume the different workplaces are independent.”

2.Typographical Corrections

oCheck “CRT Powerhas the greatest flexibility…”—spacing or punctuation is missing after “Power.”

oIn the references list, confirm consistent formatting of DOIs and parentheses.

3.Additional Implementation Details

oIn Section 3.2 (“Software Updates”), it might be helpful to briefly mention the average runtime or computational footprint for large cluster sizes (e.g., thousands of clusters). The manuscript states that GLIMMPSE 3 has no practical upper limit up to 10,000 elements. However, a short sentence regarding typical run times or memory usage would reassure readers that such calculations are feasible in practice.

4. Conclusion

This strong, timely paper describes significant enhancements to a valuable power and sample size calculation tool in linear mixed-model scenarios. The manuscript is well-grounded in inappropriate statistical theory, and the examples are engaging and informative. Incorporating the suggested revisions would further strengthen clarity—particularly regarding real-world constraints (e.g., missing data, unequal cluster sizes, or different covariance structures)—and help prospective users better understand the assumptions and usability of GLIMMPSE 3.

Recommendation: After addressing the points above, I recommend that this manuscript be accepted for publication in PLoS One. The paper makes a significant methodological contribution, and I anticipate its continued use across diverse fields requiring sound power and sample size planning for complex, clustered, and longitudinal data designs.

References and Links

•Pinheiro, J. C., & Bates, D. M. (2000). Mixed-Effects Models in S and S-PLUS. Springer. Link

•Laird, N. M., & Ware, J. H. (1982). Random-effects models for longitudinal data. Biometrics, 38(4), 963–974. Link

•GLIMMPSE 3 web application. Link

•GitHub Repositories for GLIMMPSE: Front end, Back end, pyGlimmpse.

•GNU General Public License version 2 (GPLv2). Link

(All links were accessed on January 4, 2025. The final acceptance of this manuscript will benefit from minor revisions to address these suggestions.)

Reviewer #2: 1.Add a concise, plain-language summary at the manuscript’s start to outline GLIMMPSE 3’s purpose, key features, and benefits for non-statisticians, such as clinicians designing clinical trials.

2.Simplify complex sections (e.g., Section 2) with intuitive explanations or visual aids, such as diagrams illustrating the reversible mixed model transformation. Consider moving intricate equations to an appendix to enhance readability for a general scientific audience.

3.Expand Table 1 to include additional power calculation software (e.g., PASS, G*Power, or simulation-based tools like ClusterPower) and incorporate more features, such as open-source status or handling of missing data.

4.Clearly highlight GLIMMPSE 3’s unique strengths for specific study designs, such as longitudinal studies with clustering, to differentiate it from competing tools.

5.Expand Section 7 to thoroughly address the implications of key limitations, such as the inability to manage unequal cluster sizes or missing data, and suggest practical workarounds for researchers.

6.Include details on computational performance, such as processing times for large clusters or numerical stability for high-dimensional matrices, to inform users about practical usability.

7.Provide detailed descriptions of figures (e.g., Figures 2–6) in the text, clarifying their content and relevance. For instance, explain how Figure 2’s interface simplifies specifying cluster structures.

8.Enhance Table 1 by adding more software comparisons and columns (e.g., computational approach, open-source status), and ensure all tables (e.g., Tables 3–8) have clear captions and footnotes explaining assumptions or fictional elements.

9.For each example in Section 4, explicitly identify which elements differ from the original studies and provide robust justification for input values (e.g., cite specific studies or meta-analyses for correlations and standard deviations).

10.Remove or justify fictional elements (e.g., VegF genotypes in Example 5) to strengthen the credibility and realism of the examples.

11.Simplify the methodology for identifying NIH-funded studies in Section 6.1 and clearly explain why certain funding mechanisms were excluded to improve transparency.

12.Reframe or remove the comparison to Bioconductor, emphasizing GLIMMPSE 3’s unique contributions to clinical trial and study design instead.

13.Emphasize how GLIMMPSE 3 supports researchers in designing robust clinical trials or observational studies in fields like oncology, cardiology, or public health. For example, connect Example 5 (vessel tortuosity) to its potential applications in cancer research to illustrate practical impact.

14.Discuss how GLIMMPSE 3 can be applied to observational studies or, if feasible, non-Gaussian outcomes to appeal to a diverse readership, ensuring the software’s utility across various research contexts.

Reviewer #3: Review of PONE-D-24-57373: “Power and Sample Size for Balanced Linear Mixed Models with Clustering and Longitudinality: GLIMMPSE 3”

The authors describe relevant, practical information concerning the use of GLIMMPSE 3, the latest version of GLIMMPSE, a free, web-based, open-source software tool, which calculates

power and sample size for general linear mixed models with Gaussian errors. This tool and supporting information concerning its use provide much-needed support for the research community in helping to navigate the complexities of power estimations and corresponding sample size calculations, which are fundamental not only in grant applications, but also in helping to estimate the predictive value of study results toward confirmation of tested hypotheses. The paper is very well-organized and well-written. I anticipate that this excellent work will be greatly appreciated by the research community.

I have only minor comments:

P.6. Equation 16 is actually 3 equations and Equation 17, 4 equations. For the sake of consistency perhaps it would be best to either label each equation with its own number or skip the labelling altogether for those equations.

P.9. I do not find Figure 1 in the version of the manuscript that I downloaded, only the figure caption “Figure 1: Overview of the GLIMMPSE 3 Architecture.” I presume that the authors intend to insert a figure similar to Figure 1 from their 2013 article on GLIMMPSE.

P.16. “a English” -> “an English”.

P.23. Please consider dropping the 3 from the title of section 6 in “GLIMMPSE 3 Use Statistics”: That section appears to discuss statistics for all prior versions of GLIMMPSE.

6. PLOS authors have the option to publish the peer review history of their article (what does this mean?). If published, this will include your full peer review and any attached files.

Reviewer #1: **Yes: **Osama Abdelhay

Reviewer #2: No

Reviewer #3: No

---

## [Author Response · Author response to Decision Letter 1]

8 Jul 2025

A carefully formatted 16 page response to review was uploaded via the editorial manager. Because the response includes formatted math, I am unable to respond here.

---

## [Decision Letter · Decision Letter 1]

21 Jul 2025

Power and Sample Size for Balanced Linear Mixed Models with Clustering and Longitudinality: GLIMMPSE Version 3

PONE-D-24-57373R1

Dear Dr. Glueck,

We’re pleased to inform you that your manuscript has been judged scientifically suitable for publication and will be formally accepted for publication once it meets all outstanding technical requirements.

Kind regards,

Abhik Ghosh

Academic Editor

PLOS ONE

Additional Editor Comments (optional):

Reviewers' comments:

Reviewer's Responses to Questions

**Comments to the Author**

1. If the authors have adequately addressed your comments raised in a previous round of review and you feel that this manuscript is now acceptable for publication, you may indicate that here to bypass the “Comments to the Author” section, enter your conflict of interest statement in the “Confidential to Editor” section, and submit your "Accept" recommendation.

Reviewer #2: All comments have been addressed

Reviewer #3: All comments have been addressed

2. Is the manuscript technically sound, and do the data support the conclusions?

Reviewer #2: Yes

Reviewer #3: Yes

3. Has the statistical analysis been performed appropriately and rigorously? 

Reviewer #2: Yes

Reviewer #3: Yes

4. Have the authors made all data underlying the findings in their manuscript fully available?

Reviewer #2: Yes

Reviewer #3: Yes

5. Is the manuscript presented in an intelligible fashion and written in standard English?

Reviewer #2: Yes

Reviewer #3: Yes

6. Review Comments to the Author

Reviewer #2: (No Response)

Reviewer #3: The authors have addressed my previous comments. Perhaps a new round of revisions is unnecessary, but prior to publication the authors may wish to correct the following, from line 762: "If the loss-to-followup rate is known to be x%, an adjusted sample size may be calculated by using the full data sample size, divided by the quantity (1 − x)." Surely, x here is a proportion, not a percentage, so it either needs to be defined as such or the divisor needs to be replaced by the quantity 1 - x/100.

7. PLOS authors have the option to publish the peer review history of their article (what does this mean?). If published, this will include your full peer review and any attached files.

Reviewer #2: No

Reviewer #3: No

---

## [Editor Report · Acceptance letter]

PONE-D-24-57373R1

PLOS ONE

Dear Dr. Glueck,

I'm pleased to inform you that your manuscript has been deemed suitable for publication in PLOS ONE. Congratulations! Your manuscript is now being handed over to our production team.

Kind regards,

on behalf of

Dr. Abhik Ghosh

Academic Editor

PLOS ONE